# Effect of Semi-Transverse Ventilation Velocity on Combustion Characteristics of Pool Fire Sources in a Scaled Tunnel

Liyue Gong [1], Yifan Peng [1], Jun Xu [1,*], Wanli Li [1], Tianyao Jia [2], Junqiu Ma [2] and Haihang Li [3]

1 Wenzhou Mass Transit Railway Investment Group Co., Ltd., Wenzhou 325000, China; 00002241@wzmtr.com (L.G.); 00001273@wzmtr.com (Y.P.); 00001260@wzmtr.com (W.L.)
2 Zhejiang Academy of Emergency Management Science and Technology, Hangzhou 310000, China; jty091@126.com (T.J.); mjq080913@163.com (J.M.)
3 Safety Engineering Institute, College of Quality and Safety Engineering, China Jiliang University, Hangzhou 310018, China; lihaihang@cjlu.edu.cn
* Correspondence: 00001078@wzmtr.com

**Abstract:** Compared to longitudinal ventilation, there are few studies on fire source development under semi-transverse ventilation. This work studied the influence of semi-transverse ventilation on the combustion characteristics of fire sources in a scaled tunnel. The burning rate and heat transfer feedback during pool fire combustion were revealed under different longitudinal and transverse ventilation velocities. The results showed that transverse ventilation had little influence on combustion characteristics, and the burning rate was more obviously affected by longitudinal ventilation. The heat convection feedback increased monotonically with the increase of the longitudinal ventilation, which led to the increase of the total heat feedback on the fuel. The heat radiation feedback changed little, and the heat conduction feedback decreased monotonically with the increase of the longitudinal ventilation velocity. By aid of a Fire Dynamics Simulator, it was found that the flame tilted downstream and was in the flow line of the lower cold air flow coming from upstream and the upper hot smoke flow outgoing in the downstream direction. The transverse ventilation of 2 m/s or lower hardly affected the combustion field of the fire source. Therefore, semi-transverse ventilation is preferable to longitudinal ventilation from the point of view of limiting fire expansion.

**Keywords:** semi-transverse ventilation; burning rate; heat transfer feedback

## 1. Introduction

Tunnels have brought great convenience to human travel [1,2]; however, at the same time accidents have occurred, of which fire safety issues are particularly noteworthy [3]. Smoke is the most dangerous factor in a tunnel fire [4–7]; therefore, the effectiveness of the smoke ventilation system will directly affect the safe evacuation of personnel and fire rescue operations. According to the different design requirements of the tunnel, there are three main ventilation methods: longitudinal ventilation, full transverse ventilation, and semi-transverse ventilation [8].

Longitudinal ventilation refers to the smoke exhaust mode in which the smoke flows along the tunnel. Full transverse ventilation is designed with air ducts in the tunnels for even smoke exhaust and even air make-up. Semi-transverse ventilation is similar to transverse ventilation. For semi-transverse systems, only supply vents or only exhaust vents are in operation [8]. Semi-transverse ventilation can use air ducts for smoke exhaust and tunnel entrances for air make-up. Alternatively, semi-transverse ventilation can use air ducts for air make-up and tunnel entrances for smoke exhaust. The transverse ventilation has been proven to have the best smoke exhaust effect, but the construction of the transverse ventilation system is difficult and the cost is high. Longitudinal ventilation has proven effective, but it is not suitable for two-way traffic tunnels. Therefore, the smoke ventilation mode is decided according to the actual engineering situation.

Lovas et al. [9] studied the optimization of extraction vents in tunnel transverse ventilation systems, comparing the performance of three different ventilation designs in terms of efficiency and robustness through numerical simulations. Blanchard et al. [10] found that the longitudinal ventilation rate has a significant effect on the heat release rate, smoke flow, and energy balance from experimental and numerical simulation results. Salizzoni et al. [11] investigated the relationship between the buoyancy-induced flow and the ventilation system in a tunnel ventilation system and proposed a new method to measure and scale the buoyancy-induced flow. The experimental results showed that the shape and location of the ventilation system have a great influence on the buoyancy-induced flow, whereas the density ratio and Richardson number have less influence on the buoyancy-induced flow.

Burning rate is one of the key parameters studied by researchers, and many have used pool fire sources to carry out their studies [12–18]. Blinov and Khudiakov's work [19] on pool fire is one of the most classic works; the results provided detailed references for later studies. Kang et al. [20] used n-heptane as a fuel to further investigate the effect of pool size on the combustion characteristics of a pool fire in terms of the flame height, burning rate, and fuel temperature distribution. Ping et al. [21] conducted pool fire experiments in open space using aviation fuel as the experimental material and came to the same conclusions as Kang et al. [20]. Tu et al. [22] conducted complementary experiments on larger-sized pools to investigate the change in the burning rate.

Ventilation and airflow in the tunnel are also environmental factors that need to considered. Due to the special characteristics of tunnel structures, factors affecting the burning rate of tunnel fires include the location of the fire source, ventilation velocity, burning area, and fuel type [16,17,23–27]. Naoshi Saito et al. [16] used methanol and n-heptane as fuels; the longitudinal ventilation velocity in their work ranged from 0.1 to 1.0 m/s. The results showed that, similar to the combustion of open environment pool fires, the combustion rate of pool fires in tunnels decreased with the increase of the longitudinal ventilation velocity. Li et al. [25] used pool diameter and longitudinal ventilation as variables and n-heptane as fuel and found that the burning rate of tunnel fires was significantly greater than that of open environments, and the burning rate of pool fires increased with the increase of pool size, which was similar to the findings of Blinov and Khudiakov et al. [19]. Jae Seong Roh et al. [26] conducted pool fire experiments with n-heptane in arched tunnels and found that the variation of burning rate of different sizes of pools was similar to that reported by Li et al. [25].

In summary, various factors affecting the burning rate in tunnels have been extensively studied and conclusions have been drawn. However, the vast majority of studies on the tunnel smoke exhaust mode have used longitudinal ventilation, and studies on the combustion characteristics of tunnel fire sources in transverse or semi-transverse ventilation conditions are scarce. This work analyzes the burning rate and heat feedback mechanism under semi-transverse ventilation using experimental measurement and numerical simulation. The conclusions of this work will provide a new basis for the selection of the smoke evacuation method.

## 2. Materials and Methods

### 2.1. Model Tunnel and Ventilation Mode

The Froude similarity criterion is widely used to study the flow and heat transfer of fire [8]. The model tunnel, fire power, and ventilation velocity were designed according to the Froude similarity criterion in this work. A model tunnel was built in the laboratory at a ratio of 1:15 according to the actual tunnel. As shown in Figure 1, the size of the model tunnel is 0.4 m (height) $\times$ 0.4 m (width) $\times$ 6 m (length). The front side wall of the tunnel adopted transparent fireproof glass for observing the experimental process. The rear side wall, bottom surface, and ceiling of the tunnel were made of 1 cm thick glass magnesium plate.

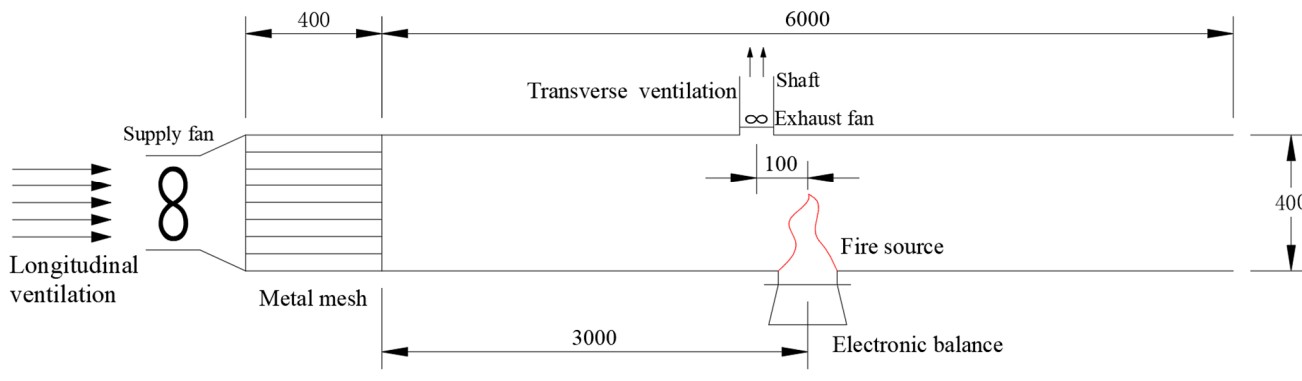

(**a**) Diagram of the model tunnel (mm)

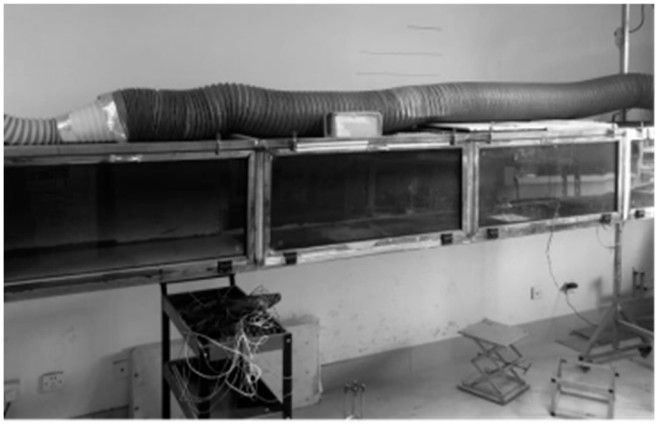

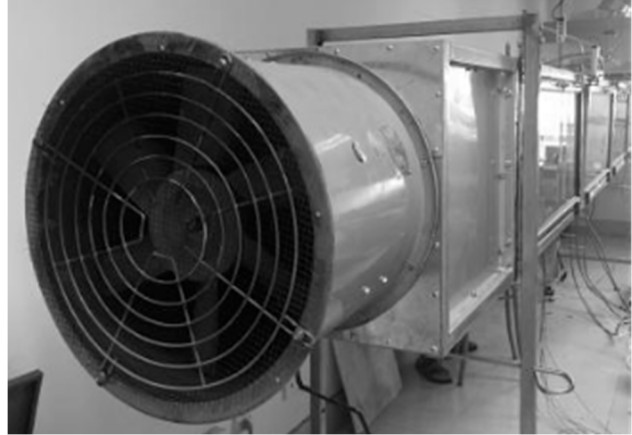

(**b**) Photo of the main body

(**c**) Photo of the longitudinal ventilation fan

**Figure 1.** Schematic diagram and photos of the model tunnel.

In our work, the use of air ducts for smoke exhaust and tunnel entrances for air make-up was employed for semi-transverse ventilation. The air make-up was achieved through longitudinal air supply, mechanical air supply in the left opening (upstream), and natural air supply in the right opening (downstream). A supply fan with adjustable velocity was installed at the left opening of the tunnel, and a porous metal mesh was used as a rectifier to ensure uniform air flow in the tunnel. The smoke exhaust was achieved by a shaft and a exhaust fan. The shaft was drilled in the tunnel ceiling at a distance of 0.1 m from the fire source, and an exhaust fan with adjustable velocity was installed in the shaft. The cross sectional dimension of the shaft was 0.1 m × 0.1 m.

According to China's current standard "Guidelines for Design of Ventilation of Highway Tunnels" [28], the longitudinal ventilation velocity shall not exceed 2 m/s for semi-transverse ventilation, and the velocity of the transverse ventilation outlet shall not exceed 10 m/s. According to the conservation of the Froude number, the maximum longitudinal ventilation velocity is 0.5 m/s, and the maximum transverse ventilation velocity is 2.5 m/s in this work. By adjusting the longitudinal fan, the velocity in the tunnel was set to 0 m/s, 0.2 m/s, and 0.4 m/s. By adjusting the frequency of the transverse fan, the velocity of the transverse vent was set from 0 to 2.0 m/s (Table 1).

**Table 1.** Experimental conditions of semi-transverse ventilation.

| Longitudinal Ventilation Velocity (m/s) | Transverse Ventilation Velocity (m/s) | Pool Size (cm) | Fuel Initial Mass (g) |
|---|---|---|---|
| 0 | 0, 0.5, 1, 1.5, 2 | | |
| 0.2 | 0, 0.5, 1, 1.5, 2 | 12 | 227 |
| 0.4 | 0, 0.5, 1, 1.5, 2 | | |

### 2.2. Iron Pool and Measuring Instrument

The fire source was located at the center axis, 3 m from the tunnel entrance. The height of the pool was adjusted so that the upper edge of the pool was flush with the tunnel floor. The square pool was 4 cm high, and the sides were 12 cm long and made of 5 mm thick iron plate. Pool fire burning is the result of the evaporation of liquids because of heat feedback [29] from the combustion volume. The heat transfer feedback on fuel determines its evaporation rate, which in turn determines its burning rate. In order to measure heat conduction feedback and heat radiation feedback during the combustion process, the pool was designed as shown in Figure 2.

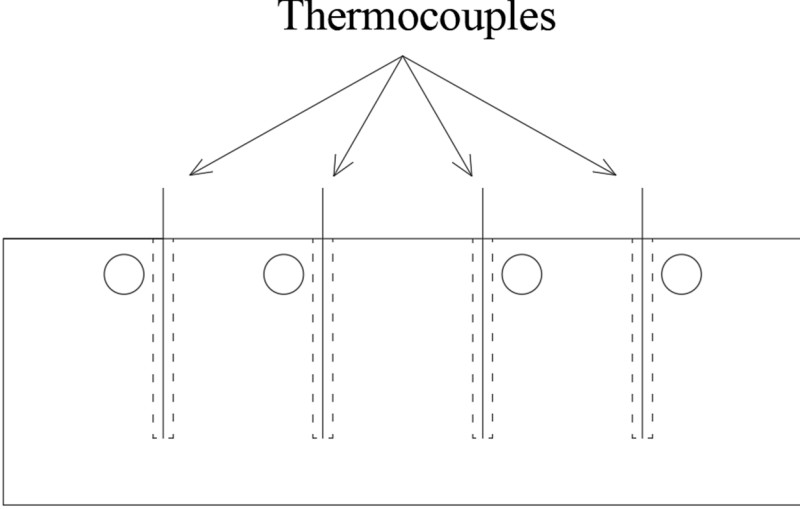

(**a**) Side view of the pool

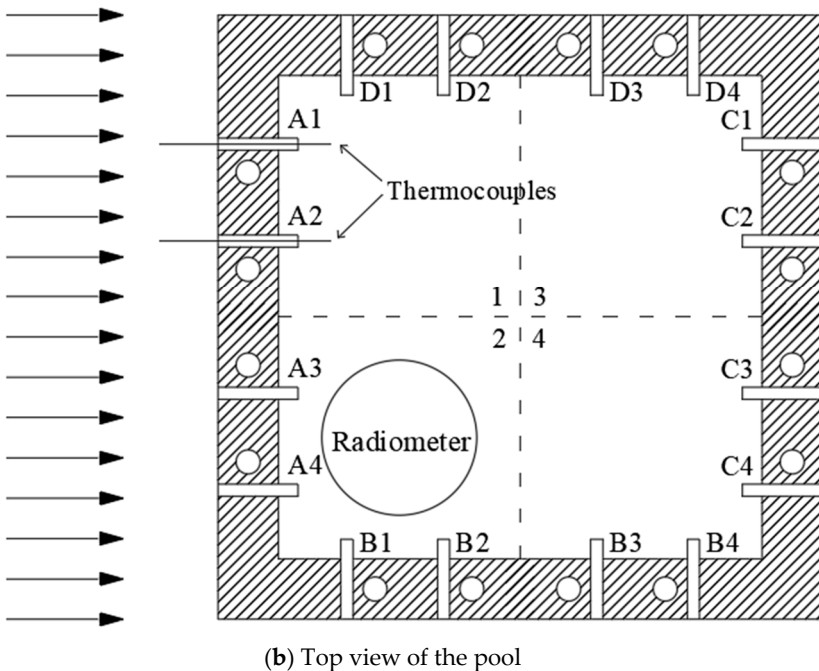

(**b**) Top view of the pool

**Figure 2.** Diagram of heat transfer feedback measurement.

The heat conduction feedback is determined by the difference between the pool wall temperature and the fuel temperature. The arrangement of thermocouples for measuring the temperature of the pool wall is shown in Figure 2a, with four pairs of thermocouples

on each side of the pool. The thermocouples were inserted into holes with a diameter of 1.5 mm and a depth of 3 cm in the side walls. The thermocouples for measuring the fuel temperature at 1 cm from the wall are shown in Figure 2b. The thermocouples entered the pool through the holes in the side walls. By processing the measured temperature values at the end of the experiment, the heat conduction feedback was calculated.

The method of measuring the heat radiation feedback values during combustion in the pool is shown in Figure 2b. The surface area of the pool was divided equally into four zones (zones 1 and 2 were windward, and zones 3 and 4 were leeward, according to the ventilation direction). In each experiment, the heat radiation feedback in one zone was measured with a radiometer. The pool was then rotated to measure the heat radiation feedback for other zones [30].

The positions of the radiator and fuel are shown in Figure 3. Quartz glass placed on top of the radiometer can effectively block damage to the radiometer by high temperature. Quartz glass has a high spectral transmission [30,31] and does not affect the measurement of the radiometer. According to the pre-test results, when the ethanol boiled and reached a steady stage, the liquid level dropped to the upper edge of the radiometer, so that the value measured by the radiometer was equal to the heat radiation feedback received on the fuel surface.

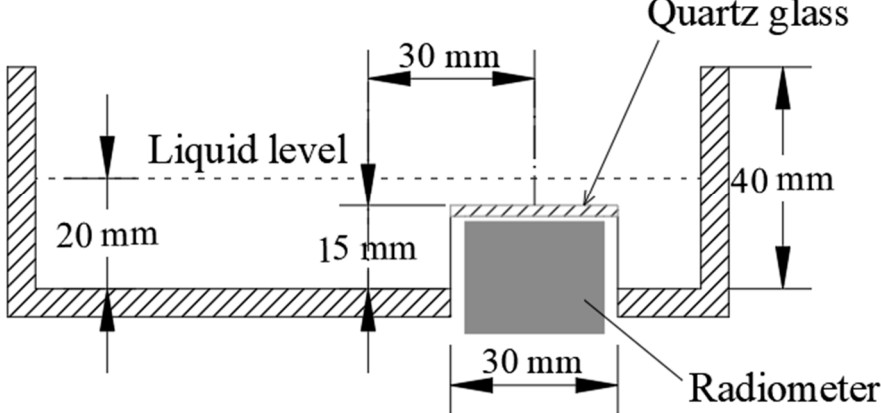

**Figure 3.** Side view of the pool.

Ethanol was used as the fuel in the experiment. The initial masses of the fuel were 227 g. It was proved that these initial weights could guarantee a sufficient steady combustion time during the experiment. The electronic balance (0.01 g precision) was located below the pool to record the transient quality of the fuel. To ensure reproducibility, two tests were performed for each condition.

### 2.3. Heat Transfer Feedback Calculation

The burning rate is mainly controlled by the heat conduction feedback, heat radiation feedback, and heat convection feedback. In order to analyze the change pattern of combustion, heat transfer during combustion should be analyzed. Heat conduction feedback and heat radiation feedback can be calculated by recording changes in the readings of the thermocouples and radiometer, respectively.

### 2.3.1. Heat Conduction Feedback

When a pool fire burns, a portion of the heat of the flame is transferred to the pool wall, which in turn transfers the heat to the fuel through heat conduction. By measuring the pool wall temperature and the fuel temperature, the value of the heat conduction feedback can be calculated.

Figure 4 shows the change in the A1 surface pool wall temperature and fuel temperature when the longitudinal ventilation velocity is 0.2 m/s and the transverse ventilation velocity is 0 m/s. Analyzing the change rule of temperature, it could be concluded that

the combustion was divided into three stages: the initial stage, the steady stage, and the extinction stage. In the steady stage, the ethanol reached the boiling state, its mass loss rate tended to stabilize, and the pool wall temperature and fuel temperature also tended to stabilize. It should be noted that in the extinction stage, the liquid level dropped to near the bottom of the pool, and the flame height was reduced substantially. The thermocouple entering the pool through the small hole in the side wall of the pool was no longer below the liquid level but was in direct contact with the flame, so the thermocouple temperature value increased steeply in the extinction stage.

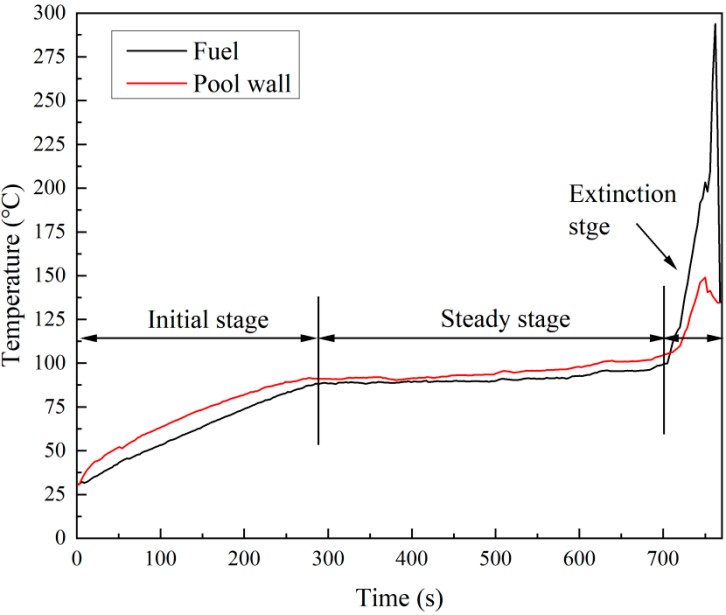

**Figure 4.** Temperature changes of pool wall and fuel.

Temperature values for the steady stage were calculated by taking the average value. The temperature of the pool wall, as shown in Figure 4, was 89.58 °C, and the temperature of the fuel was 95.24 °C. The heat conduction feedback was calculated by using the temperature difference between the pool wall and the fuel.

The value of heat conduction feedback is calculated by Equation (1) [32]:

$$\dot{q}''_{cond} = h(T_w - T_l)$$

$$= \begin{cases} 0.59\dfrac{k_l}{L^{\frac{1}{4}}}\left[\dfrac{g\beta}{v\alpha}\right]^{\frac{1}{4}}\left(T_w - T_l\right)^{\frac{5}{4}}, 10^4 \leq Ra_L \leq 10^9 \\ 0.1k_l\left[\dfrac{g\beta}{v\alpha}\right]^{\frac{1}{3}}\left(T_w - T_l\right)^{\frac{4}{3}}, 10^9 \leq Ra_L \leq 10^{13} \end{cases} \qquad (1)$$

where $L$—fuel depth (0.015 m);

$g$—gravity acceleration (9.8 m/s$^2$);
$k_l$—thermal conductivity (W/m°C);
$\beta$—heat expansion coefficient (1/°C);
$v$—kinematic viscosity (m$^2$/s);
$\alpha$—heat diffusion coefficient (m$^2$/s);
$T_w$—pool wall temperature (°C);
$T_l$—fuel temperature (°C).

The segmentation function is determined by the magnitude of the Rayleigh number RaL, which can be calculated using Equation (2):

$$Ra_L = P_r \cdot G_r = \frac{g\beta L^3(T_w - T_l)}{v\alpha} \qquad (2)$$

where $Ra_L$—Rayleigh number;

$G_r$—Grashof number;

$P_r$—Prandtl number.

We calculated the temperature difference and heat conduction of the 16 pairs of thermocouples (A1-D4) on the four walls of the pool and summed the thermal feedbacks using Equation (3) to obtain the total heat conduction feedback $\dot{q}_{cond}$:

$$\dot{q}_{cond} = \frac{\Sigma_{i=A1}^{D4} \dot{q}''_{cond} Ld}{4} \tag{3}$$

where $d$—pool side length (0.12 m).

### 2.3.2. Heat Radiation Feedback

Figure 5 shows the change of the leeward surface without ventilation (longitudinal and transverse ventilation velocities were 0 m/s). As can be seen from Figure 5, the heat radiation feedback entered the steady stage at approximately the same time as the temperature and reached stability when the ethanol reached the boiling state. The values for the steady stage were calculated using the averaging method. The radiometer measures a voltage signal, which needs to be converted to derive the value of the heat radiation feedback. The heat radiation feedback, as shown in Figure 5, was about 3.71 kW/m$^2$.

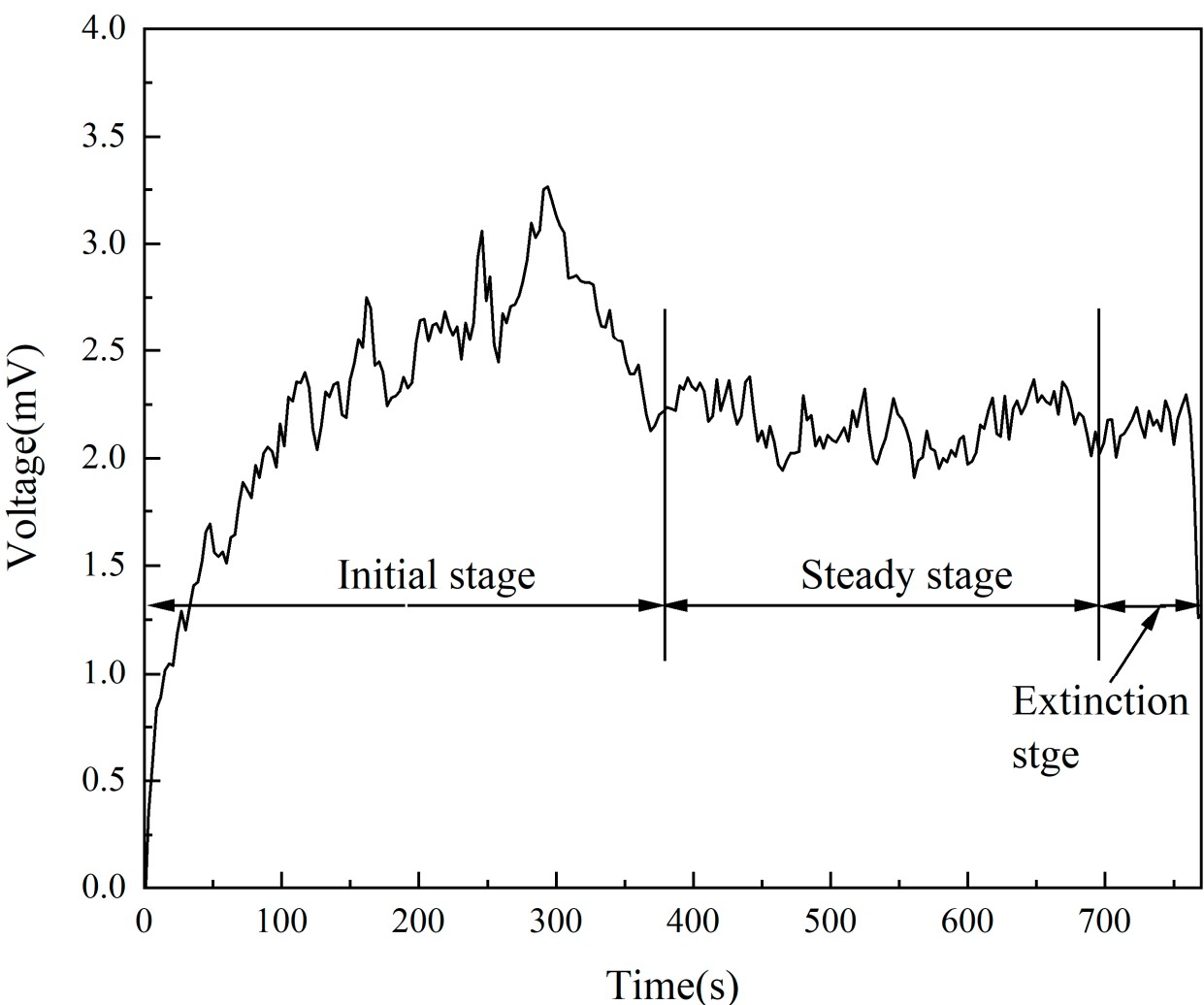

**Figure 5.** Radiation measurement of leeward surface without ventilation.

After obtaining the heat radiation feedback value measured by the radiometer, the total heat radiation feedback $\dot{q}_{rad}$ can be calculated by Equation (4) [32]:

$$\dot{q}_{rad} = \Sigma_{i=1}^{n} \dot{q}_{rad}'' \cdot S_i \tag{4}$$

where $S_i$—windward or leeward surface area (m$^2$);

$\dot{q}_{rad}''$—radiation value measured by the radiometer (kW/m$^2$).

### 2.3.3. Heat Convection Feedback

The amount of heat $Q$ required for the fuel to evaporate in the pool fire can be calculated by Equation (5) [32]:

$$\dot{Q} = \dot{m} \left[ C_p (T_{boil} - T_0) + h_{fg} \right] \tag{5}$$

where $\dot{m}$—burning rate (kg/s);

$C_p$—specific heat capacity (kJ/kg·K);
$T_{boil}$—fuel boiling temperature (°C);
$T_0$—initial temperature (25 °C);
$h_{fg}$—evaporation latent heat (kJ/kg).

The proportion of heat conduction feedback $\chi_{cond}$ to total heat transfer feedback can be calculated by Equations (3) and (5):

$$\chi_{cond} = \dot{q}_{cond}/\dot{Q} = \frac{\Sigma_{i=A1}^{D4} \dot{q}_{cond}'' Ld/4}{\dot{m} \left[ C_p (T_{boil} - T_0) + h_{fg} \right]} \tag{6}$$

The proportion of heat radiation feedback $\chi_{rad}$ to total heat transfer feedback can be calculated by Equations (4) and (5):

$$\chi_{rad} = \dot{q}_{cond}/\dot{Q} = \frac{\Sigma_{i=1}^{n} \dot{q}_{rad}'' \bullet S_i}{\dot{m} \left[ C_p (T_{boil} - T_0) + h_{fg} \right]} \tag{7}$$

The heat convection feedback $\dot{q}_{conv}$ in the pool fire can be calculated according to Equations (3)–(5):

$$\dot{q}_{conv} = \dot{Q} - \left( \dot{q}_{cond} + \dot{q}_{rad} \right) \tag{8}$$

The proportion of heat convection feedback $\chi_{conv}$ to total heat transfer feedback can be calculated by Equations (6) and (7):

$$\chi_{conv} = 1 - \left( \chi_{cond} + \chi_{rad} \right) \tag{9}$$

According to the calculated heat convection feedback, the convective heat transfer coefficient $h_c$ can be calculated by Equation (10) [33]:

$$h_c = \frac{\dot{q}_{conv} \cdot B}{A \left[ In(1+B) \right] (T_R - T_{boil})} \tag{10}$$

where $A$—pool area (m$^2$);

$B$—Spalding B number;
$T_R$—fuel evaporation temperature (°C);
$T_{boil}$—fuel boiling temperature (°C).

## 3. Results and Discussion

### 3.1. Changes in the Burning Rate

The burning rate is the most intuitive physical quantity one can use to observe the combustion characteristics of a pool fire [14,15]. The decisive factor affecting the burning rate is heat transfer [34,35]. The burning rate of the ethanol pool fire in the horizontal tunnel under semi-transverse ventilation is shown in Figure 6. Two main results can be drawn from this figure: (1) The burning rate of the pool fire was basically unchanged with the increase of the transverse ventilation velocity. Transverse ventilation did not change the air flow field near the pool fire, so it basically had no impact on the pool fire. (2) The burning rate of the pool fire was affected by longitudinal ventilation. The burning rate of the pool fire increased monotonically with the increase of longitudinal ventilation velocity. This was because the heat convection feedback accounted for a relatively large proportion of the heat transfer in the case of longitudinal ventilation. Longitudinal ventilation changed the convection form, enhanced the heat convection feedback, and then increased the burning rate of the pool fire.

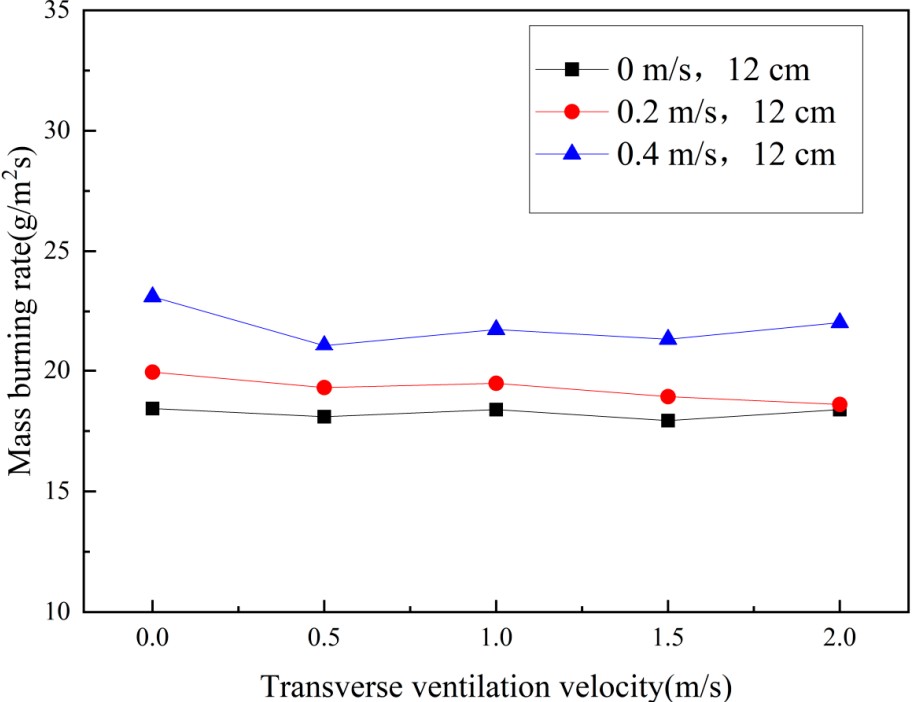

**Figure 6.** Mass burning rate under semi-transverse ventilation condition.

There is a consensus that heat transfer processes alter the pool fire burning rate, which can be summarized as follows: When a smaller longitudinal ventilation velocity is present, flame tilting occurs, resulting in a decrease in the heat radiation feedback from the flame to the fuel [30]. When a larger longitudinal ventilation velocity exists, flame downwash occurs on the leeward side of the pool [36], enhancing the heat transfer between the pool wall and the fuel [37]. Changes in heat conduction, heat radiation, and heat convection feedback will be analyzed next to reveal the changes in the pool fire burning rate.

### 3.2. Heat Conduction Feedback Analysis

The average wall temperature and the average fuel temperature change of the pool in the horizontal tunnel under different ventilation velocities is shown in Figure 7. Figure 7 shows that the average temperature of the wall and fuel was less affected by transverse ventilation and more affected by longitudinal ventilation.

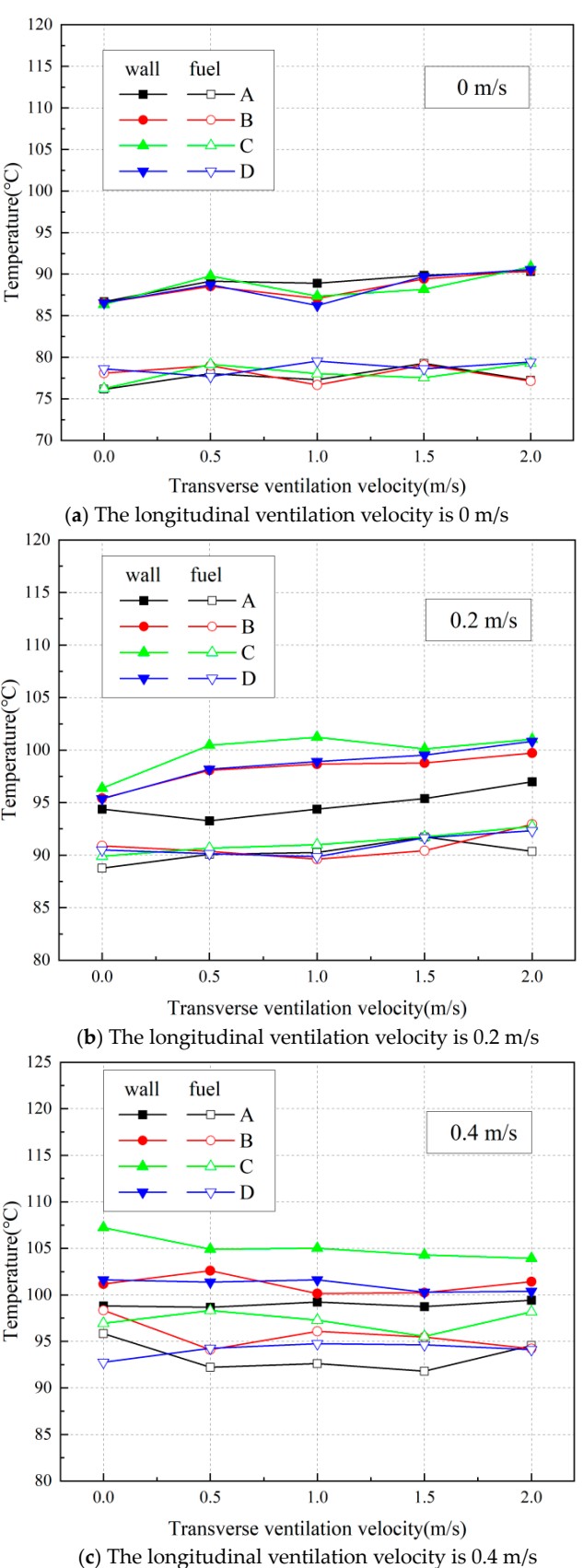

(**a**) The longitudinal ventilation velocity is 0 m/s

(**b**) The longitudinal ventilation velocity is 0.2 m/s

(**c**) The longitudinal ventilation velocity is 0.4 m/s

**Figure 7.** Pool wall and fuel temperature.

In the presence of longitudinal ventilation, the average temperature of the walls and fuel increased monotonically with the longitudinal ventilation velocity. The heat of the

wall mainly comes from the flame. As the longitudinal ventilation velocity increases, the fuel receives more air and burns more completely. The flame transfers more heat to the wall of the pool, which then increases the average temperature of the pool wall. For the average temperature of the fuel, because the heat of the ethanol fuel evaporation mainly comes from heat convection feedback, the increase in longitudinal ventilation enhances the heat convection feedback, so the average temperature of the fuel increases accordingly.

The average temperature of the pool wall and the fuel increased as a whole, but the range of variation on the four wall sides is quite different. The windward side (side A) had the lowest temperature and the smallest increase, because the cooling effect of longitudinal ventilation reduced the overall temperature of the windward side, and the windward side received the least heat from the flame because the flame tilted leeward [27,38]. In contrast, the leeward side (side C) had the highest temperature and the largest increase, because the flame deflected near the leeward wall and the fuel, and the leeward side (side C) received more heat from the flame. The temperature values on the parallel side of the pool (side B and side D) were the same as the overall growth trend but not as high as those on the leeward side (side C).

The heat transfer feedback was calculated using Equations (1)–(3). With the change in ventilation velocity, the change in heat conduction feedback was determined, as shown in Figure 8. As can be seen from Figure 8, the change in heat conduction feedback was similar to the trend of the burning rate. The influence of transverse ventilation on heat conduction feedback was small, and the influence of longitudinal ventilation was more obvious.

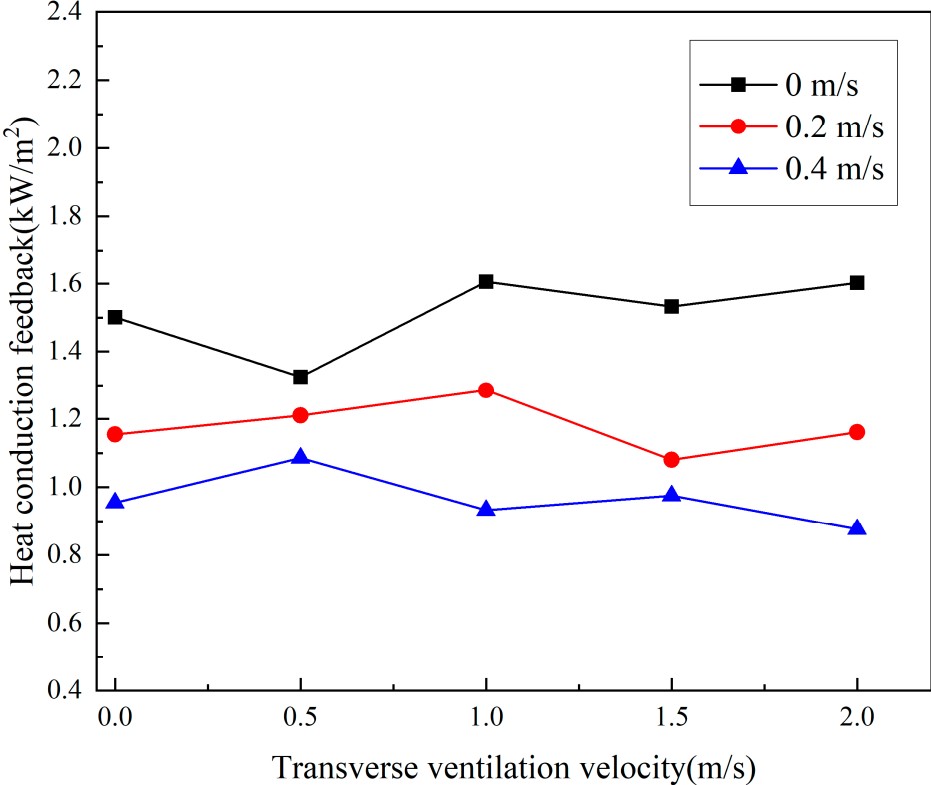

**Figure 8.** Heat conduction feedback.

In the presence of longitudinal ventilation, the heat conduction feedback monotonically decreased as the longitudinal ventilation velocity increased. This indicated that the increase in fuel temperature was larger than that of the wall temperature, reducing the temperature difference between the wall and the fuel. This change can be seen in Figure 7.

### 3.3. Heat Radiation Feedback Analysis

The heat radiation feedback of the pool fire was calculated using Equation (4). The heat radiation change of the windward and leeward surface of the pool fire in the horizontal tunnel under different ventilation velocities is shown in Figure 9. As can be seen from Figure 9, the change trend of the heat radiation feedback with ventilation velocity was similar to those of mass burning rate and heat conduction feedback, which were little affected by transverse ventilation and more affected by longitudinal ventilation.

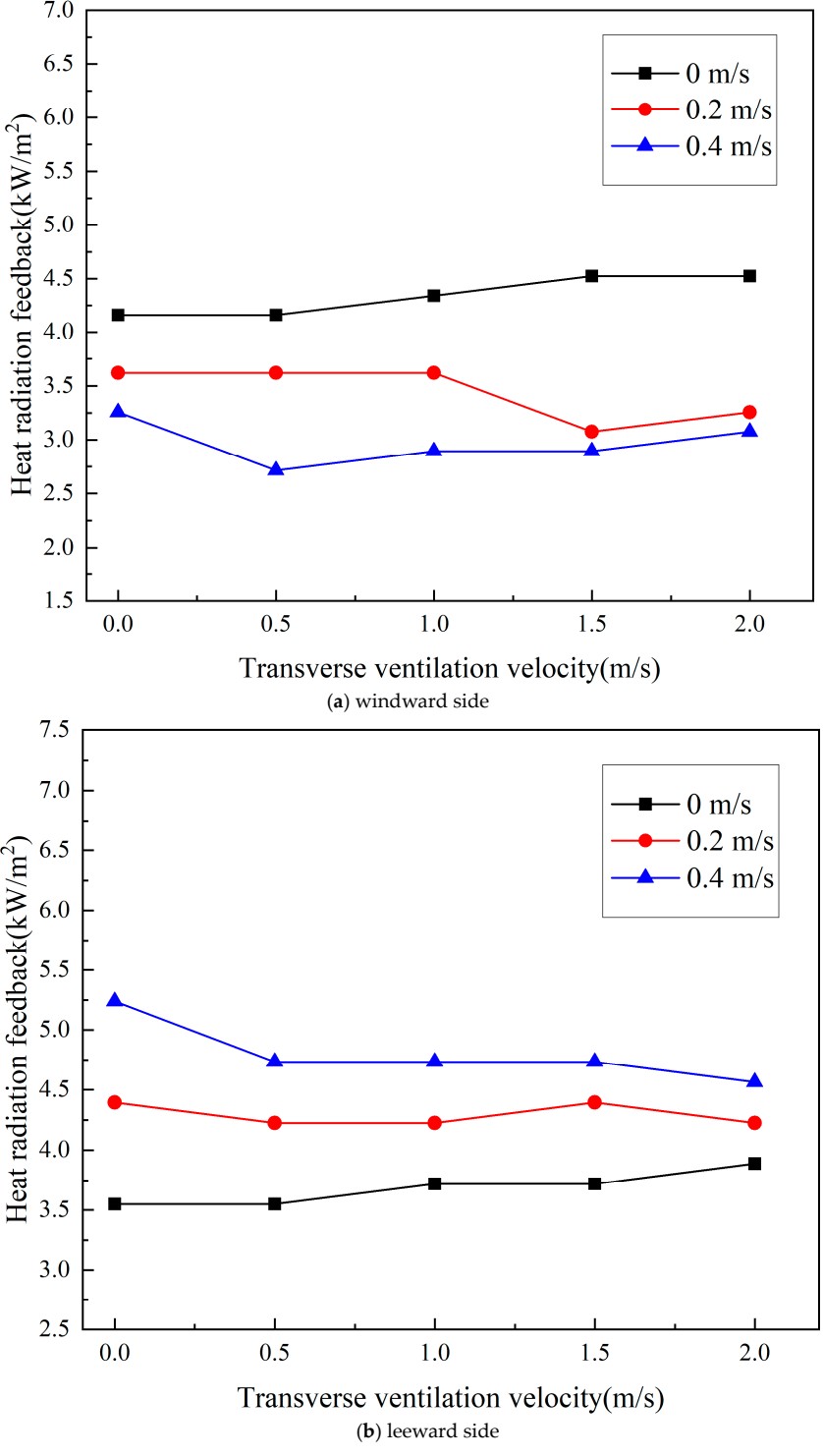

**Figure 9.** Heat radiation feedback on windward and leeward sides.

The heat radiation feedback on the windward side decreased monotonically with the increase of the longitudinal ventilation velocity, while the change on the leeward side was just the opposite. This is because, in the presence of longitudinal ventilation, the flame is tilted near the leeward side, and the heat radiation from the flame received by the leeward side increases. With the increase of the longitudinal velocity, the flame is increasingly tilted and the heat radiation received by the leeward side is higher, but the windward side is just the opposite.

The total heat radiation feedback change of the pool fire in the horizontal tunnel under different ventilation velocities is shown in Figure 10. As can be seen from Figure 10, the total heat radiation feedback change of the pool fires was not large with the change of ventilation velocity.

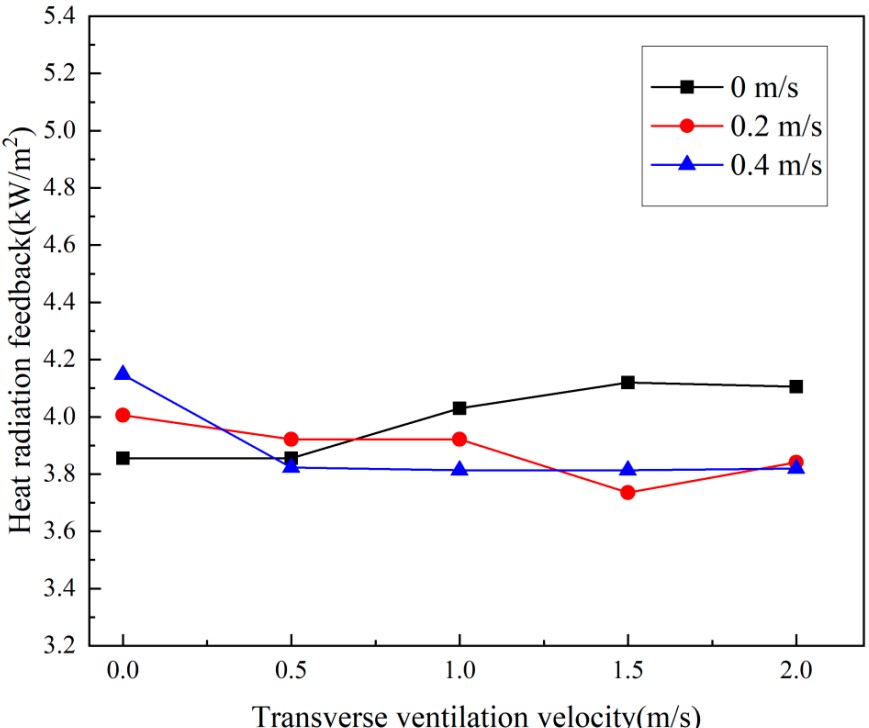

**Figure 10.** Co-occurrence of institutional cooperation.

*3.4. Heat Convection Feedback Analysis*

The proportions of heat conduction, radiation, and convection feedbacks in the horizontal tunnel under different ventilation velocities are shown in Figure 11. As can be seen from Figure 11 since transverse ventilation has little impact on fire sources, the ratio of heat conduction, convection, and radiation feedback did not change much with the increase of transverse ventilation velocity.

At the transverse ventilation velocity of 0 m/s, the proportions of heat conduction, heat radiation, and heat convection feedbacks and the convection heat transfer coefficient in the horizontal tunnel are shown in Table 2. As can be seen from the table, with the increase of longitudinal ventilation velocity, the total heat feedback of the pool fire increased, the proportions of heat conduction and heat radiation feedbacks decreased, and the proportion of heat convection feedback increased. (1) The change in the proportion of heat convection feedback was most obvious, mainly because of the dominance of heat convection feedback in ethanol pool fires. Longitudinal ventilation changes the form of convection from natural convection to forced convection, which enhances the intensity of heat transfer in the combustion of the pool fire, leading to an increase in total thermal feedback and an increase in the proportion of heat convection. (2) For heat radiation feedback, because the change of heat radiation was small, the proportion of heat radiation feedback decreased with the

increase of total heat feedback. (3) Compared with heat radiation feedback, the decreased ratio of heat conduction feedback was more obvious. This was due to the fact that the heat conduction feedback decreased with increasing longitudinal ventilation velocity, while the total heat feedback increased, so the proportion of heat conduction feedback decreased.

**Table 2.** Proportions of the three heat transfer modes and convective heat transfer coefficient.

| Longitudinal Ventilation Velocity (m/s) | Total Heat Feedback (W) | $\chi_{cond}$ | $\chi_{rad}$ | $\chi_{conv}$ | $h_c$ (W/m$^2$·K) |
|:---:|:---:|:---:|:---:|:---:|:---:|
| 0 | 262.22 | 8.24% | 21.17% | 70.59% | 45.19 |
| 0.2 | 283.75 | 4.84% | 20.33% | 74.83% | 51.84 |
| 0.4 | 328.81 | 4.18% | 18.60% | 77.22% | 61.99 |

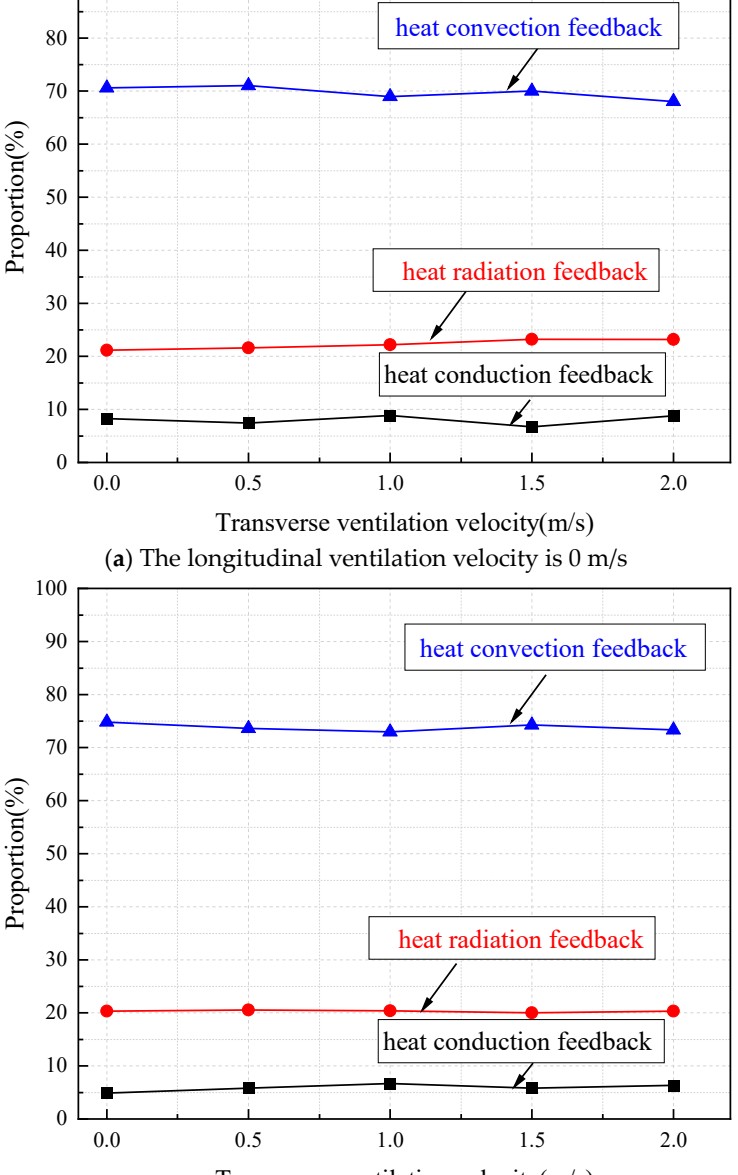

(**a**) The longitudinal ventilation velocity is 0 m/s

(**b**) The longitudinal ventilation velocity is 0.2 m/s

**Figure 11.** *Cont.*

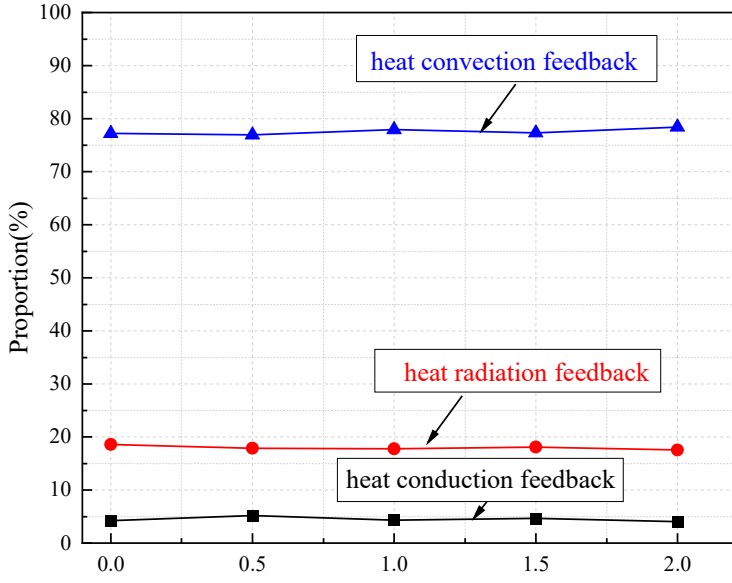

(**c**) The longitudinal ventilation velocity is 0.4 m/s

**Figure 11.** Heat conduction, heat radiation, and heat convection feedback ratios.

### 3.5. Temperature and Velocity Field Analysis

In order to provide a deeper explanation of the pool fire burning rate and heat feedback mechanism, and to further reveal the role of longitudinal and transverse ventilation in the pool fire, the numerical simulation method was employed. Figures 12 and 13 show the temperature and velocity fields (heat release rate equal to the corresponding experimental condition) derived from simulations with a Fire Dynamics Simulator (FDS). It is recognized that the fire source characteristic length should be greater than four times the grid size [39]. The calculated grid size should be no greater than 0.035 m, and the final selected grid size should also pass the independence test. The grid size of 0.03 m was selected finally by carrying out simulations with 0.035 m, 0.03 m, and 0.025 m grid sizes for comparison.

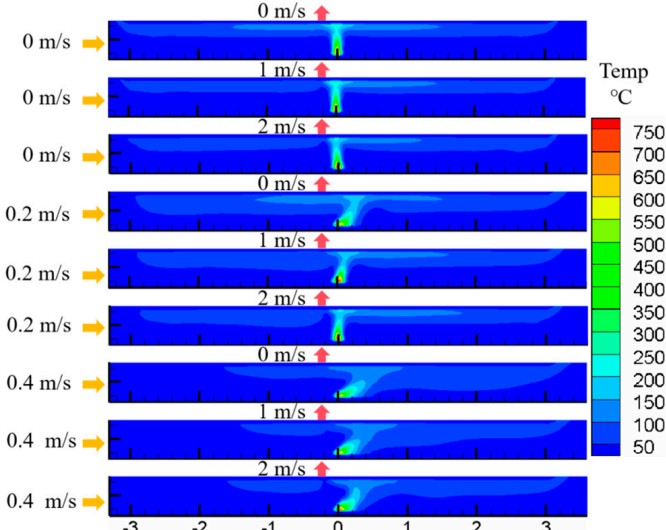

**Figure 12.** Longitudinal slice of average temperature field.

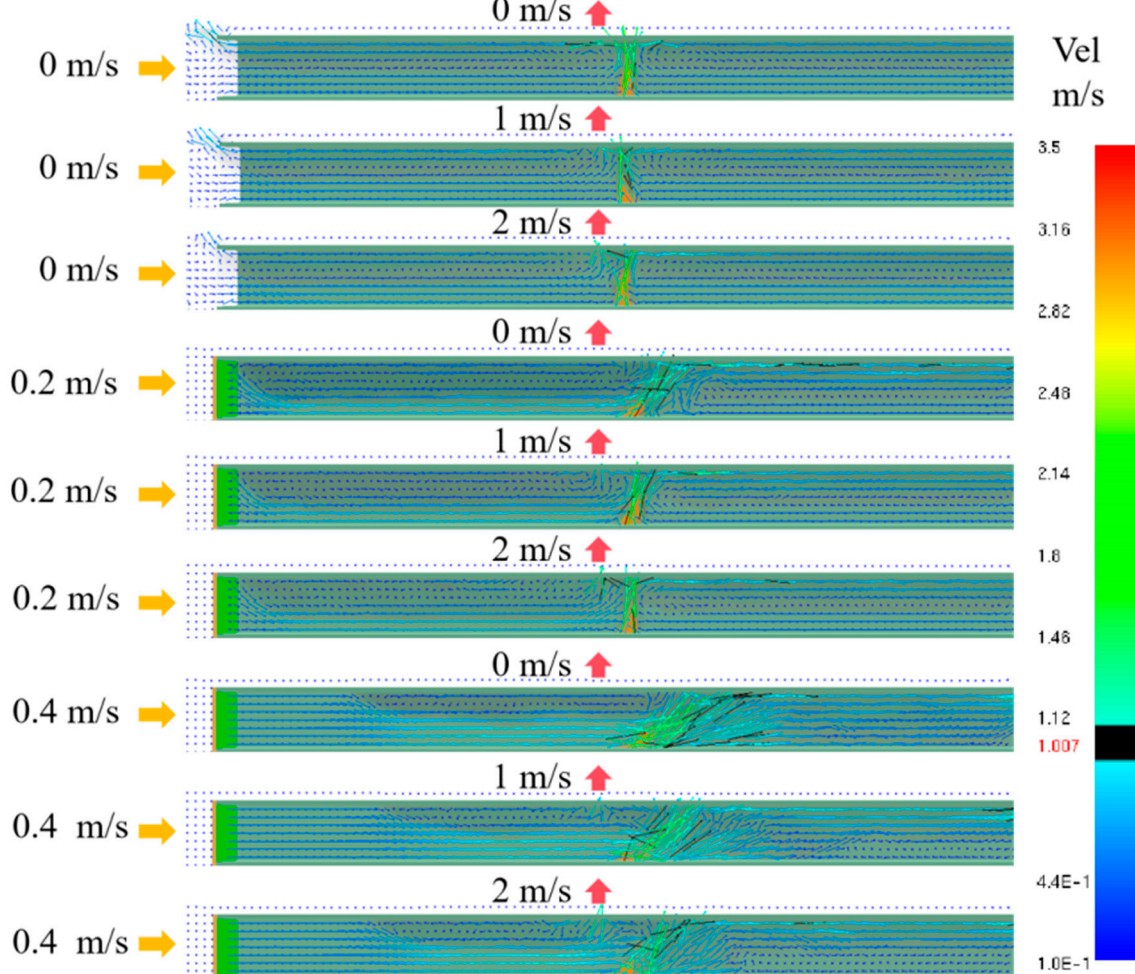

**Figure 13.** Longitudinal slice of instantaneous velocity field and smoke layer.

Figure 12 shows longitudinal slices of the mean temperature field in the steady state for different cases. It can be found that (1) the overall temperature distributions of the fire plumes were similar, with the average flame temperatures all being about 500 °C and the average smoke layer temperatures all being about 150 °C; (2) as the longitudinal ventilation velocity increased, the flame tilted downstream. The smoke backlayering length in the upstream decreased, and the smoke layer thickness first increased and then decreased. The smoke layer thickness in the downstream increased, and more smoke was blown downstream and discharged from the right opening; and (3) with the increase of transverse ventilation velocity, the temperature and smoke layer thickness in the upstream decreased slightly, and the smoke backlayering length remained almost the same. The temperature and smoke layer thickness in the downstream were almost constant.

Based on the above results, it can be concluded that although the transverse ventilation velocity (0.5–2 m/s) was much larger than the longitudinal ventilation velocity (0.2–0.4 m/s), the former had a much smaller effect on the flame and smoke layer than the latter.

The effect of transverse ventilation on the flame can be reflected by the velocity field. Figure 13 shows the instantaneous velocity field and smoke layer during the steady state. From Figure 13, it can be seen that (1) the fire plume rose at about 3 m/s under all ventilation conditions, which was greater than the maximum transverse ventilation velocity (2 m/s), so the significant airflow field generated by the pool fire combustion weakened the effect of transverse ventilation; (2) when the longitudinal ventilation velocity was 0 m/s (i.e., when both the left and right opening were naturally ventilated), the movement velocities

of the smoke layer to both openings were less than 1 m/s; and (3) when the longitudinal ventilation velocity was 0.2 m/s or 0.4 m/s (i.e., when the left opening was force-ventilated, and the right opening was naturally ventilated), the movement velocity of the smoke layer to the downstream was greater than 1 m/s.

Based on these results, it can be summarized that the transverse ventilation of 2 m/s or lower only changed the local air flow directly below the shaft and hardly affected the combustion field of the fire source. In particular, when there was a longitudinal ventilation air velocity, the flame tilted downstream and was in the flow line of the lower cold air flow coming from upstream and upper hot smoke flow outgoing in the downstream direction. Therefore, the effect of longitudinal ventilation on the burning rate and heat feedback mechanism was much greater than that of transverse ventilation, which could also explain the results of Sections 3.1–3.4.

## 4. Conclusions

The mass loss and the temperatures of the pool wall and fuel were measured for an ethanol pool fire in a horizontal tunnel by varying the longitudinal and transverse ventilation velocities. The burning rate as well as the values of heat conduction, heat radiation, and heat convection feedback were calculated to reveal the effect of different ventilation methods on the pool fire. The main conclusions were as follows:

(1) Transverse ventilation had little impact on the fire source in the horizontal tunnel, and the key parameters, such as burning rate, did not change with the change of transverse ventilation velocity.

(2) The fire source in the horizontal tunnel was more obviously affected by the longitudinal ventilation. For the combustion of the ethanol pool, the burning rate increased monotonically with the increase of longitudinal ventilation velocity.

(3) The heat convection feedback of the pool fire increased monotonically with the increase of longitudinal ventilation, which led to the increase of the total heat feedback; the proportion of heat convection feedback also increased.

(4) The heat conduction feedback decreased monotonically with the increase of longitudinal ventilation, and its proportion also decreased.

(5) With the increase of longitudinal ventilation, the heat radiation feedback on the windward side decreased monotonically, and the heat radiation feedback on the leeward side increased monotonically. The total heat radiation feedback did not change much, and its proportion was relatively reduced.

Transverse ventilation has little effect on the burning intensity of pool fires in horizontal tunnels, while longitudinal ventilation offers a greater contribution to the burning intensity of pool fires. Based on this conclusion, semi-transverse ventilation is recommended for smoke extraction in tunnel fires, which can effectively avoid the effect of ventilation leading to fire expansion. These conclusions are mainly based on current experimental data, and conditions and may not be applicable to all situations. In practical applications, other factors such as fire type, fuel characteristics, and changes in ventilation conditions need to be considered. In order to gain a more comprehensive understanding of the effects of ventilation on pool fire combustion, further experimental studies are needed in the future to collect more data and to consider other possible influencing factors. For example, attempts should be made to explore the effect of the location, shape, and higher ventilation velocity of transverse ventilation systems on the combustion characteristics of pool fires.

**Author Contributions:** L.G.: conceptualization, resources, and project administration; Y.P.: validation and formal analysis; J.X.: methodology, supervision, and project administration; W.L.: visualization and validation; T.J.: validation and funding acquisition; J.M.: data curation and writing—original draft preparation; H.L.: funding acquisition and writing—review and editing. All authors have read and agreed to the published version of the manuscript.

**Funding:** This research was funded by the Opening Fund of the State Key Laboratory of Fire Science (Grant No. HZ2022-KF02), Zhejiang Provincial Natural Science Foundation of China (Grant No. LGF21E040001), and the Basic Scientific Research Fund of Zhejiang Province (No. 2022YW18).

**Institutional Review Board Statement:** Not applicable.

**Informed Consent Statement:** Not applicable.

**Data Availability Statement:** All data, results, and models used or generated during the study appear in the submitted article.

**Conflicts of Interest:** Author L.G., Y.P. and W.L. were employed by Wenzhou Mass Transit Railway Investment Group Co., Ltd. The remaining authors declare that the research was conducted in the absence of any commercial or financial relationships that could be construed as a potential conflict of interest. The Wenzhou Mass Transit Railway Investment Group Co., Ltd. had no role in the design of the study; in the collection, analyses, or interpretation of data; in the writing of the manuscript, or in the decision to publish the results.

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
