# Peer review of "Effect of Semi-Transverse Ventilation Velocity on Combustion Characteristics of Pool Fire Sources in a Scaled Tunnel"

_fire, doi:10.3390/fire7010018_

Round 1

Reviewer 1 Report

Comments and Suggestions for Authors

Burning rate and heat feedback mechanism of tunnel pool fire under semi-transverse ventilation

The manuscript has the potential for acceptance after revision. To enhance the quality of the manuscript, the following suggestions are provided:

1. The title and abstract need to be modified to clearly reflect the primary purpose of the research. The authors should ensure that the title and abstract rationalize the research’s objectives and highlight its novelty. The abstract should focus on the problems addressed and the methods employed.

2. It is necessary to correct errors and mistakes, especially grammar errors, throughout the entire paper. A thorough check should be conducted.

Hint:

Line 202: “Figure 4.3”; “kW/m2”

Line 212: Pay attention to the use of regular and italic fonts. The authors should check all the formulas.

Line 214: “kJ/kg℃”

Line 215: “oil” (If it should be corrected to ethanol.)

Line 233: “m2”

3. Clearly define variables and concepts, and explain symbols when they are first introduced. Maintain a consistent font throughout.

4. Modify the introduction to provide a comprehensive overview of related investigations. Clearly state the significance of unresolved issues and the corresponding solutions proposed in the research.

5. A clear definition of the term “semi-transverse” should be provided.

6. Line 173: The definition of feedback should be provided, along with an explanation of the author’s rationale for using this specific term.

7. Line 244: According to Figure 1, the transverse vent is located not far from the fire source. Furthermore, the transverse ventilation velocity exceeds that of the longitudinal ventilation. Given these conditions, why did the transverse ventilation not alter the airflow field near the pool fire? The author should offer a logical explanation.

8. Based on Figures 6 to 11, the impact of transverse ventilation on fire rate and heat transfer appears to be minimal. Additionally, the discussion is predominantly centered around longitudinal ventilation. This raises questions about the main objectives of the research: What is the significance of incorporating transverse ventilation? If the aim is to highlight the negligible effect of transverse ventilation, this may point to a lack of innovation. The authors are advised to thoroughly reevaluate the research framework and its primary goals to ensure clarity.

9. The authors are encouraged to elucidate the significance of their heat transfer analysis. Specifically, a clear description of the relationship between the burning rate and heat transfer should be provided to enhance the understanding of these interconnected phenomena.

Comments on the Quality of English Language

The language quality should be improved.

Reviewer 2 Report

Comments and Suggestions for Authors

The effects of longitudinal ventilation and semi-transverse smoke exhaust on the combustion of the pool fire were studied in this paper, the following problems need to consider carefully by the author:

1.     For the prediction of the total heat feedback to the pool, whether Eq. (5) is right? What’s relation with the heat release rate of the fire?

2.     The influence of longitudinal ventilation on the combustion of pool fire is very complicated. Only limited experimental conditions were analyzed, the author should give the applicable range of relative conclusions.

Reviewer 3 Report

Comments and Suggestions for Authors

This paper presents an interesting experimental work, which is worth being published. However, it suffers a few weaknesses which could be corrected relatively easily to get a much more useful work. The paper falls into 4 sections. Section 1 is the introduction. After a few general sentences on fire in tunnels and smoke removal strategies, it cites a few references on fire pools and burning rate evaluations. It would be better to open a little bit more the bibliography to papers from other geographic areas, e.g. Europe for example Grant et al Proc Roy Soc 1998, PIARC data, recent work by Salizzoni’s group (e.g Sallizoni et al, Measurements and scaling of buoyancy-induced flows in ventilated tunnels, Flow, 2023 and references therein) and paper from ISAVFT conference series (many papers are of interest, e.g. Lovas et at, Optimizing the repartition of extraction vents in transverse ventilation. ISAVT 14, BHRG, 2011; Blanchard et al. Energy balance in a tunnel fire – midscale tests and CFD simulations. ISAVT 14, BHRG, 2011). Section 2 present the experimental set up and measurement methods. Experiments are based on a reduced scale tunnel 1/15 with semi transverse ventilation system. Subsection 2.1 is a general description; subsection 2.2 concentrates on the pool and its instrumentation; subsection 2.3 presents heat transfer feedback. In this subsection, detail calculations are given but there is no physical discussion of prevailing effect nor of what is new compared to other work, this could be improved, while the detail of some calculations could be omitted. Table 1 (conditions of experiments) could be improved by putting for each 8 cases considered, the main non dimensional parameters and possibly a few results. The term “transverse ventilation velocity” is ambiguous. It is written “According to China's current standard "Guidelines for Design of Ventilation of 96 Highway Tunnels"[25], the longitudinal ventilation velocity shall not exceed 2 m/s for semi-transverse ventilation, and the velocity of the transverse ventilation outlet shall not 98 exceed 10 m/s. According to the conservation of Froude number, the maximum longitu-99 dinal ventilation velocity is 0.5 m/s, and the maximum transverse ventilation velocity is 100 2.5 m/s in this work.” I deduce from this that “transverse ventilation velocity” is in fact “velocity of the transverse ventilation outlet”. Please clarify this and use another term than “transverse ventilation velocity”. Section 3 presents the results of the study. The figures are not very impressive, in the sense that they do not make any clear and unambiguous support of conclusions, with very flat curves giving the feeling that no parameter is really influent! Moreover, and this is a point which would deserve a thorough discussion, in what the results would be unaffected by a slight variation of the shape of the extraction vent? A full study would be, in my sense, to use the very good experimental presented here in conjunction with numerical simulation on other geometries, to make a full paper. In the absence of such study, the main results of the paper are in table 2. However there should be a discussion on the influence of scaling for real case application, as radiation terms are known to be prone of poor scalability due to the T^4 term in Black body radiative law. A solution for such a discussion is, again, to perform numerical simulations at scale 1/15 validated against experiment, and then full scale simulations to compare with small scale results. If the authors are not willing to perform numerical simulations in the present work, more detail should be given on geometry and experimental conditions so that the paper could be used as starting point for further study.

Round 2

Reviewer 1 Report

Comments and Suggestions for Authors

Since the authors responded to the reviewer’s questions and modified part of the article, the reviewer recommends accepting this article for publication.

Comments on the Quality of English Language

minor editing required

Reviewer 3 Report

Comments and Suggestions for Authors

Most of the comment made in the first review are taken into account.